# Classification of Sea Ice Types in Sentinel-1 SAR Data Using Convolutional Neural Networks

**Hugo Boulze** [1,*] , **Anton Korosov** [2] **and Julien Brajard** [2]

1   École Nationale des Sciences Géographiques, Univ. Gustave Eiffel, 77455 Marne-la-Vallée, France
2   Nansen Environmental and Remote Sensing Center, 5006 Bergen, Norway; anton.korosov@nersc.no (A.K.); julien.brajard@nersc.no (J.B.)
*   Correspondence: hugo.boulze@ensg.eu

**Abstract:** A new algorithm for classification of sea ice types on Sentinel-1 Synthetic Aperture Radar (SAR) data using a convolutional neural network (CNN) is presented. The CNN is trained on reference ice charts produced by human experts and compared with an existing machine learning algorithm based on texture features and random forest classifier. The CNN is trained on two datasets in 2018 and 2020 for retrieval of four classes: ice free, young ice, first-year ice and old ice. The accuracy of our classification is 90.5% for the 2018-dataset and 91.6% for the 2020-dataset. The uncertainty is a bit higher for young ice (85%/76% accuracy in 2018/2020) and first-year ice (86%/84% accuracy in 2018/2020). Our algorithm outperforms the existing random forest product for each ice type. It has also proved to be more efficient in computing time and less sensitive to the noise in SAR data. The code is publicly available.

**Keywords:** convolutional neural network; Sentinel-1; SAR; sea ice type; ice chart; Arctic

## 1. Introduction

For secure navigation and offshore activities, sea ice concentration and type in polar regions should be monitored and forecasted with high accuracy. Ice concentration is defined as the ratio of the pixel area covered by floating sea ice to to the total area. Ice type can be defined in terms of stage of sea ice development—from newly frozen smooth ice (Nilas), to deformed and roughened Young Ice, to thick ice cover that survived summer melts (Old Ice) with several intermediate stages [1]. Ice type is linked to several ice characteristics including ice thickness, surface roughness, mechanical properties and is arguably very important characteristic not only for navigation purposes but also for assimilation into numerical models.

Data from space-borne Synthetic Aperture Radar (SAR) available, for example, from the Copernicus mission Sentinel-1 belonging to the European Spatial Agency (ESA) [2] provides invaluable information both in bad weather conditions and during polar nights [3]. Typically SAR images are analyzed by ice analysts in operational centers for manual classification of ice types and drawing of ice charts. With the availability of more than 100 Sentinel-1 SAR images in the Arctic ocean per day this procedure understandingly requires a significant effort and human power.

In several recent studies automated algorithms have been developed for dual-polarization C-band SAR image segmentation and ice/water classification [4–10]; for retrieval of ice concentration [11,12]; for classification of several ice types [13–15]. It is emphasized in these studies that a threshold based classification of ice types on single frequency dual-polarization SAR data is not possible and more advanced methods employing either full polarization, or multi-frequency, or texture analysis are required.

The latter method [15] is based on Haralick texture features and random forests classifier. Twelve texture features are computed from a Gray Level Co-occurence Matrix constructed from sub-images (32 × 32 pixels) in a sliding window. A random forest classifier with 11 trees, 8 levels and 10 features was used for classification of 4 or 5 ice classes. Despite the high accuracy of the texture feature-based algorithm, it is quite complex, requires high computational power and shows strong sensitivity to thermal and texture noise in Sentinel-1 SAR data [16,17].

Deep learning through application of convolutional neural networks (CNN) to SAR images has been successfully used for ice-water classification and to estimate sea ice concentration [18,19]. These studies, however, did not attempt to distinguish different ice types on SAR images and applied CNNs to classify an entire sub-image (45 × 45 pixel in [18]) or pixel-by-pixel classification of relatively large sub-image (250 × 250 pixels, in [19]).

The main goal of our research is to develop a CNN-based algorithm for the classification of sea ice types on dual-polarization SAR data from Sentinel-1. The algorithm should combine relative simplicity and speed of the CNN approach, high accuracy and ability to classify sea ice types and to have relatively low sensitivity to noise in SAR data. Several CNNs were trained on the same dataset as in [15] for producing comparable results and for sensitivity experiments. Another CNN was trained on recent data from 2020 for operational sea ice type retrieval.

The paper is structured as follows. Section 2 describes data used for training the CNNs. Section 3 describes the CNN architecture and training process. Section 4 presents the results of training and ice type classification. Section 5 sheds light on experiments with hyperparameter tuning and CNN sensitivity. Section 6 discusses advantages of CNN ice charts and aspects of operational application.

## 2. Data and Preprocessing

The study area stretches from the Canadian Arctic Archipelago (90°W) to Franz Josef Land (53°E) and from 85°N to 75°N. It includes the Lincoln Sea, the Fram Strait, the Greenland Sea, the Barents Sea and Svalbard (see figures in Section 4.2). Two sources of data are used in our study: satellite data are used to feed the classification algorithm and expert data are used for training and validation. The data are split into two datasets: one dataset from 2018 (hereafter referred to as 2018-dataset) is the same dataset as in [15] and is used for sensitivity experiments and training a CNN for comparison with the results from [15]. Another dataset from 2020 (2020-dataset) is used for building a CNN for operational sea ice type retrieval. The data details and preprocessing steps are further described below.

### 2.1. Satellite Data

Sentinel-1 SAR data acquired in Extra Wide Swath mode with medium resolution in HH/HV polarization were downloaded from Copernicus Open Access Hub (https://scihub.copernicus.eu). Each scene covers approximately 400 × 400 km with pixel size of 40 m. Each pixel contains values of normalized radar cross section (radar backscatter intensity, denoted as $\sigma^0$) in two polarizations: HH (both emitted and received electromagnetic pulses are horizontally polarized) and HV (emitted in horizontal polarization, received in vertical polarization). Radar backscatter depends on dielectric properties and roughness of the scattering surface (here, sea ice or water). Measurements in two polarizations (HH and HV) can be used to better differentiate between surface types using both the signal intensity and texture [15] (see Figure 1).

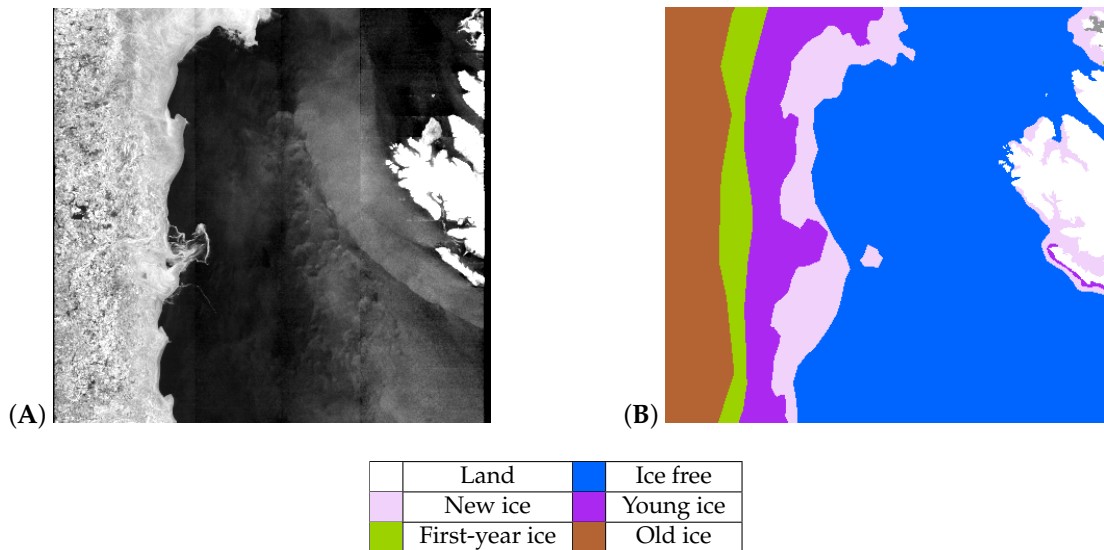

| | | | |
|---|---|---|---|
| | Land | | Ice free |
| | New ice | | Young ice |
| | First-year ice | | Old ice |

**Figure 1.** Example of a Sentinel-1 Synthetic Aperture Radar (SAR) image in HV polarization acquired on 19-Feb-2018 at 07:19:44 (**A**) and a corresponding manual ice chart showing stage of ice development (ice type) (**B**). Values in the table indicate the standard World Meteorological Organization (WMO) code for these ice types. Size of SAR images here and below is 400 × 400 km.

The 2018-dataset contains 44 SAR scenes acquired between January 2018 and March 2018. It was used for experimenting with the CNN hyperparameters and training of a neural network for comparison with the results from [15]. These scenes were the same scenes as used by [15], and were pre-processed with thermal and textural noise correction and incidence angle correction as described in detail in [15–17]. Values of $\sigma^0$ were converted from dB to 255 gray levels in the range $[-31, 0]$ dB for HH, $[-32, -7]$ dB for HV and saved with 8-bits unsigned integer precision.

The 2020-datasets contained 1500 Sentinel-1 SAR scenes acquired in January–February 2020. Only 255 scenes were manually selected by visual comparison with the expert data for training and validating the CNN. Visual comparison allows to identify scenes that match the best with the ice chart and were, presumably, used by an ice expert for building an ice chart (see details below). An improved texture noise removal procedure was applied to satellite data in this dataset. As explained in [17], texture noise is manifested as increased variance of $\sigma^0$ proportional to the thermal noise induced by the SAR antenna. As well as the thermal noise, the texture noise is especially pronounced when the signal is low (e.g., in areas with calm ocean surface). The new texture noise correction was implemented as follows:

$$\sigma^0_C = \frac{\sigma^0 * w_1 + G(\sigma^0) * w_2}{w_1 + w_2} \tag{1}$$

where $\sigma^0$ denotes radar backscatter corrected for thermal noise, $G$—Gaussian filter, $w_1$ is the weight of original signal and $w_2$ is the weight of smoothed signal. Our experiments showed that optimal results are achieved when the Gaussian filter has window size of 3, $w_1$ is equal to 0.5 and $w_2$ is equal to signal to noise ratio (SNR) computed as follows:

$$w_2 = \frac{G(\sigma^0)}{\sigma^0_N} \tag{2}$$

where $\sigma^0_N$ is thermal noise equivalent backscatter. For more details on thermal and texture noise impact see [16,17], as further explanations fall outside the scope of this paper. For practical reasons, values of corrected $\sigma^0$ were saved with 32-bits floating point precision.

## 2.2. Expert Data

The expert data includes sea ice charts provided by the National Ice Center (NIC, https://www.natice.noaa.gov/Main_Products.html). A sea ice chart is a vector map in shape-file format operationally prepared by an ice expert based on available SAR, optical and passive microwave satellite imagery. The ice chart contains polygons with three different types of labels for each polygon: total sea ice concentration indicates the area fraction covered by sea ice of whichever type; stage of development (SoD) indicates the dominant type of sea ice; and the partial ice concentration indicates the concentration of the dominant ice type. Detailed ice charts are provided by NIC every three days and are prepared by ice experts based on latest available SAR images from Sentinel-1, Radarsat-2, passive microwave and optical data. Information about data sources is not presented on ice charts.

The ice charts are rasterized following the approach suggested in [15] with ice chart pixel of 1 km. As a result, the raster ice charts have the same geometry as the corresponding SAR images but lower resolution (see Figure 1) and the pixels contain SoD labels—ice type codes accepted in the Ice Chart Color Code Standard by the World Meteorological Organization (WMO) [1]. Not all standard ice types are possible to automatically separate on SAR imagery. For example, such ice types as first-year medium ice and first-year thick ice are almost identical and are rarely separated on ice charts. In our experiments only four ice class types are used:

1. Ice free (water) (WMO codes: 1)
2. Young ice (WMO codes: 83–85)
3. First-year ice (WMO codes: 87–94)
4. Old ice (WMO codes: 95–97)

## 2.3. Preparation of a Combined Dataset

Combination of SAR data and ice charts for CNN training/testing includes several steps explained below.

First, an extra layer is added to the ice chart—distance to border (*D*). Each pixel in this layer contains a value of distance to the nearest border between two ice types. This value is afterwards used for filtering valid records. Under assumption that the borders on the manually drawn ice charts are not very detailed, only pixels that are far enough from the border are considered as valid.

Second, another extra layer is added to the ice chart—manual mask of invalid pixels. The manual mask is added in order to avoid inclusion of pixels with incorrect values of $\sigma^0$ visible on SAR images. Some SAR images contain thermal noise or radio frequency interference artifacts that can affect the CNN results (see Figure 2). These features are manually identified on SAR images and ice charts are edited in image manipulation software by replacing concerned pixel with white color (see Figure 3). In addition, some of the pixels with obviously wrong classification by ice expert are also blanked out including cases when the border between ice types on manual ice chart is coarse and a polygon of one ice type obviously includes ice of a different type (see examples in Section 6 and Figure 9 in [15]). Although this manual correction has to be done only once—during the training phase, it is worth noting that it is very time consuming and was performed only for the 2018-dataset.

Next, a SAR image is divided into samples of square sub-images of the same size thus generating an input data cube of size ($N \times K \times K \times 2$), where *N*—denotes total number of samples, *K*—sub-image size, 2—two layers of one sub-image from HH and HV polarizations. The sub-image size is kept relatively small and one sub-image corresponds to one label from the ice chart. In our study $K = 50$ as explained in Section 5.2.

SoD labels are taken from the corresponding pixels of the rasterized ice chart thus forming the output 1D vector **c** with size *N*. The SoD labels are one-hot encoded [20]—each label $c_i$ is replaced with a binary vector $\mathbf{y}_i$ with length equal to the number of classes *C* (in our case, $C = 4$). The *k*-th element of $\mathbf{y}_i$ is set to 1 and other elements—to 0, where *k* equals SoD label. For example, $\mathbf{y}_i = [1, 0, 0, 0]$ corresponds to $c_i = 1$ (ice free) and $\mathbf{y}_i = [0, 1, 0, 0]$ corresponds to $c_i = 2$ (young ice).

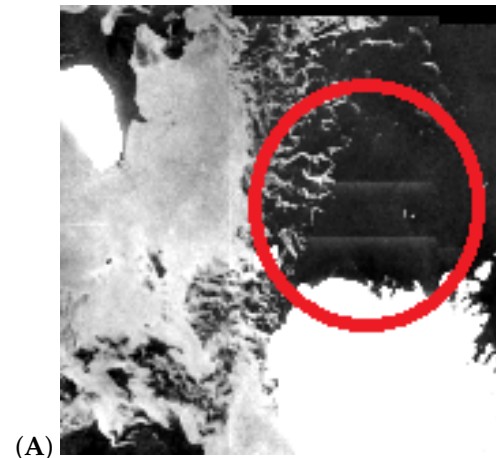
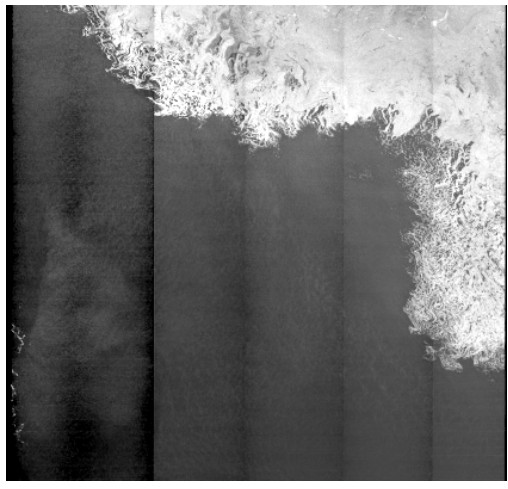

**(A)**  **(B)**

**Figure 2.** Examples of pattern of radio frequency interference (whitish horizontal stripes inside the red circle, (**A**) and inter-swath boundary (**B**) on a Sentinel-1 SAR images (respectively on 23-Jan-2018 at 6:06:29 and 26-Feb-2018 at 03:51:54).

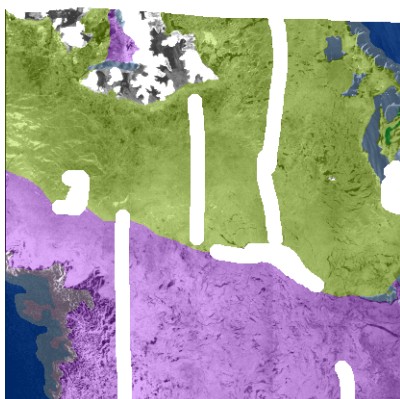

**Figure 3.** An example of manual mask of invalid pixels (white color) used to mark invalid pixels on the ice chart. Semitransparent colors on the foreground show young-ice, first-year ice and ice free areas from an ice chart. Gray-scale image on the background is a Sentinel-1 SAR image (26-Feb-2018 at 4:44:09).

For the 2020-dataset yet another layer was added to the ice chart for filtering out invalid input data—CNN segment size. As shown in the results (see figures in Section 4.1) the CNN produces very patchy ice charts with many segments often as small as just 1 pixel. Obviously, such small details cannot be compared with the rather generalized manual ice charts. The "CNN segment size" layer contains size of individual segments produced by a CNN. For generating such layer, first, a preliminary CNN is trained on all data and applied to all scenes in the training dataset. This CNN produces a preliminary classification of ice types (Figure 4A). Next, the neighbor pixels on the preliminary classification with the same ice type are grouped into segments and the size of each segment is calculated (Figure 4B). Finally, the pixels that belong to segments with too small size are marked as invalid (Figure 4C).

Values of distance to border, manual invalid mask (only for 2018-dataset) and CNN segment size (only for the 2020-dataset) are also taken from the corresponding pixels and form 1D vectors with size $N$. These vectors are used to define valid data from the input and output datasets.

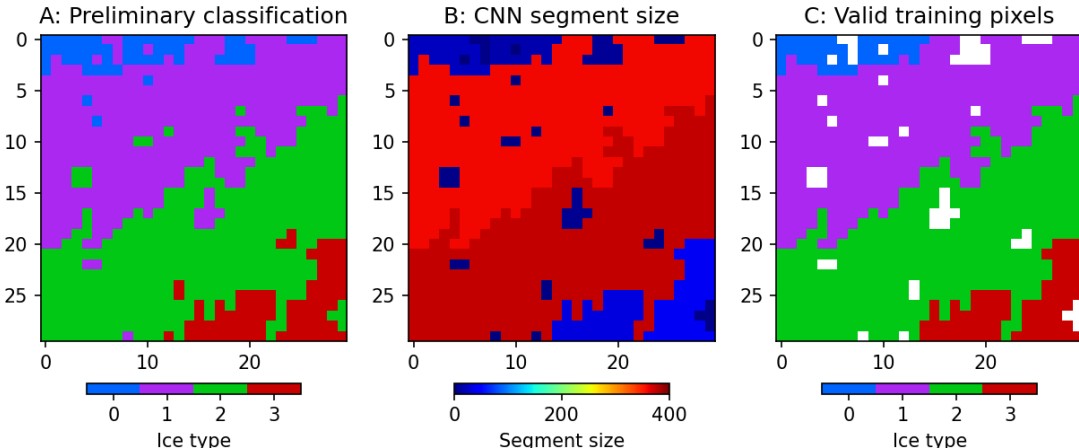

**Figure 4.** An example of convolutional neural network (CNN) segment size computation. (**A**)—classification results obtained with a preliminary CNN. (**B**)—size of segments obtained with the preliminary CNN. (**C**)—valid pixels that were selected for further training (invalid pixels are marked with white color).

Finally, the dataset is randomly permuted and then split into the training and validation subsets using 70/30 ratio with the same proportion of ice types in both parts. The validation dataset-2018 is used to evaluate the sensitivity to parameters described in Section 5. In this case, even though the common practice is to use a third independent test dataset to evaluate a model, the final score is presented on the same validation dataset in order to compare with performance obtained in Park et al. [15] where no test dataset is used. On the other hand, the validation dataset-2020 is not used to tune the method and marginally used during the training, so it constitutes a valid dataset to evaluate the final score of the method.

The sub-datasets are randomly permuted again and fed into the training algorithm as described below.

## 3. CNN Architecture and Training

### 3.1. CNN Architecture

The classifier used is a multi-layered feed-forward neural network called "convolutional neural network" (CNN): each layer takes as an input the previous layer output (except the first layer which takes satellite images as input), and calculates its own output (see hereafter how this calculation is performed).

The CNN is composed of 2 batch-norm layers, 3 convolutional layers, 2 max-pooling layers, 3 hidden dense layers, 4 dropout layers (used only for the training) and one output layer. Figure 5 presents the details about the particular sequence of layers ("architecture") and hyperparameters of each layer (i.e., the parameters of the layer that are not determined in the training) for the selected configuration.

Note that the sensitivity to some hyperparameters, the architecture and the setting of the training, are evaluated in Section 5.1.

After convolutional layers and dense layers (except for the last layer), a non-linear function is applied to the computed values, so the CNN can handle a non-linear link between the images in input and the ice type in output. The non-linear function chosen in our setup is the rectifier linear unit (ReLU) [21] defined as

$$f(x) = \max(0, x), \tag{3}$$

where $x \in \mathbb{R}$.

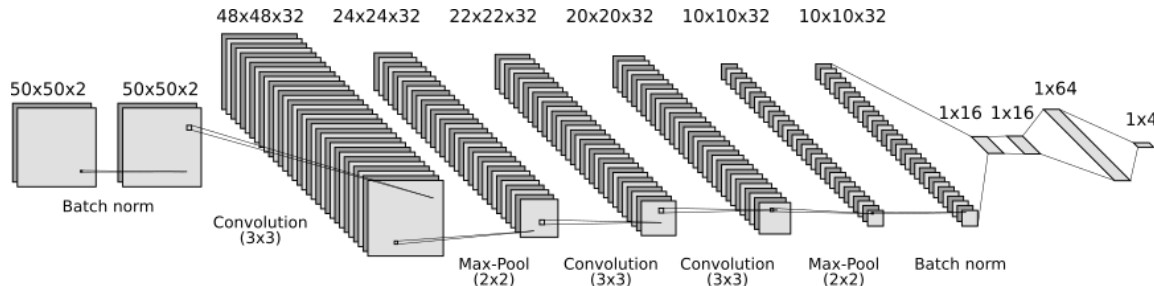

**Figure 5.** The CNN architecture. The dense layers are the four last layers (on the right). The dropout layers (not represented in the figure) are inserted before each dense layer. The figure was generated using http://alexlenail.me/NN-SVG/LeNet.html.

Batch-norm layers rescale the output of the previous layer so that the mean value is close to 0 and the standard deviation is close to 1. They have two trainable parameters, one for the offset, one for the rescaling, for each input feature. They are used to facilitate the training, and have also a regularization effect. For more details see Ioffe and Szegedy [22].

Convolutional layers perform a convolution by a kernel of $3 \times 3$ and apply the non-linear ReLU function, defined in Equation (3). The values of the kernel are the parameters to be determined during the training phase. Convolutional layers have proved their efficiency to extract predictive features from image data [23].

Max-Pooling layers decrease the size of the intermediate features (output of convolutional layers) by selecting only the max value for each patch of $2 \times 2$ pixels, so the size of each feature after max-pooling is divided by $2 \times 2 = 4$.

Dense layers are also called "fully connected layers". Each neuron of the layer takes as an input a linear combination of the previous layer's outputs, and then applies the non-linear ReLU function, defined in Equation (3). The weights of the linear combination are the parameters to be determined during the training phase.

Dropout layers disable randomly, for each computation, a portion $p_D$ of the output of the previous layers by setting them to zeros. These layers are activated only during the training phase whereas when the neural network is used in inference (to determine an ice type from an input image), they have no effect. The effect of dropout layers is to regularize the training to avoid overfitting [24].

Finally the output layer is a particular type of dense layer. The layer contains 4 neurons corresponding to the 4 classes to be determined by the CNN. The activation function used is a softmax function, it computes the output $p_k$ as

$$p_k = \frac{e^{z_k}}{\sum_{j=1}^{C} e^{z_j}}, \tag{4}$$

where $e$ is the exponential function, $p_k (1 \leq k \leq C)$ is the $k$-th output of the layer and $\mathbf{z} = \{z_1, \cdots, z_C\}$ is the result of the $C$ linear combinations of the previous layer's outputs.

The outputs are then all included between 0 and 1, and their sum is equal to one, so they can be interpreted as a probability of belonging to each class. The output with the highest probability is selected as the ice type.

*3.2. CNN Training*

Convolutional and dense layers contain some parameters (also called "weights") that are determined through an iterative optimization process of minimizing the loss

$$L(\boldsymbol{\theta}) = -\frac{1}{C} \sum_{i=1}^{n} \sum_{k=1}^{C} y_{ik} \log(p_{ik}) + \gamma \frac{1}{N_{\boldsymbol{\theta}}} \sum_{j=1}^{N_{\boldsymbol{\theta}}} \theta_j^2 \tag{5}$$

where $\boldsymbol{\theta}$ is the vector of size $N_\theta$ stacking all the parameters to be determined from all the layers, $i$ is the index of the sample and $k$ the index of the class. $p_{ik}$ is the label predicted by the CNN for a sample $i$, see Equation (4). $y_{ik}$ is the observed label one-hot encoded (see Section 2.3) and log is the logarithm function. The second term of the loss is a L2-regularization term weighted by $\gamma$ aiming at avoiding overfitting by penalizing high values of parameters. In our setting, $N_\theta$ = 72,008 and we recall that, $C = 4$. The loss function is optimized iteratively on batches of $n$ samples (corresponding to $n$ sub-images of size $50 \times 50$) using the Adam [25] optimizer. The batches are taken randomly from the training dataset, Adam [25] is a gradient-based optimizer of stochastic loss functions (i.e., in case where $n$ in Equation (5) is smaller than the size of the training dataset). The optimization is performed over several epochs, one epoch being achieved when the entire dataset has been presented to the neural network for optimization. The two most sensitive parameters of the optimizers are the batch size $n$ and the learning rate which is an internal parameter of the Adam optimizer. To avoid overfitting, dropout layers are activated with a dropout rate of $p_D > 0$ and the L2-regularization parameter $\gamma$, see Equation (5), is strictly positive. Table 1 gives all the parameters used in our setup. A sensitivity study of the training parameters is presented in Section 5.1.

**Table 1.** The hyperparameters used for training the final CNNs.

| Learning Rate | Dropout rate $p_D$ | Batch Size $n$ | L2 Regularization Parameter $\gamma$ |
|---|---|---|---|
| 0.001 | 0.1 | 512 | 0.001 |

The CNN for the 2018-dataset (CNN-2018) was trained in 28 epochs on $180 \times 10^3$ samples and validated on $45 \times 10^3$ samples selected from 44 Sentinel-1 SAR images. After epoch 20 the validation accuracy was not increasing. CNN-2018 was produced with the TensorFlow v1.13 and Keras 2.2.4 libraries. The CNN code, weights and example of usage are available on GitHub repository [26].

The CNN for 2020-dataset (CNN-2020) was trained using TensorFlow v2.1 and Keras v2.2.4-tf libraries in 30 epochs on $840 \times 10^3$ samples and validated on $360 \times 10^3$ samples selected from 255 images. Samples were split between "ice free", "young ice", "first-year ice" and "old ice" classes in the proportion 2:1:1:2. The validation accuracy stopped significantly increasing after epoch 20. Training was performed on computers with configuration specified in Table 2.

**Table 2.** Configuration of computers for CNN training.

| CNN Name | OS | CPU | RAM, GB | GPU |
|---|---|---|---|---|
| CNN-2018 | Windows 10 | Intel Core i7-7500U $2 \times 2.7$ GHz | 12 | NVIDIA GeForce 940MX |
| CNN-2020 | Ubuntu 18.4 | AMD Ryzen 2990WX $32 \times 3.5$ GHz | 128 | N/A |

*3.3. Validation Metrics*

To validate the results of our algorithm the following metrics on the validation dataset are computed:

- The confusion matrix: An element of the matrix at the row $r$ and column $c$ is the number of samples predicted in the class $r$ over the number of samples in the class $c$. A perfect classification would lead to a diagonal confusion matrix with 1 on the diagonal and 0 elsewhere. Each column sums up to 1.
- The accuracy per class (diagonal of confusion matrix) is the ratio of the correctly predicted samples of a given class over the total number of samples of the same class.
- The overall accuracy is the ratio of the correctly predicted samples over all the samples in the validation set. Overall accuracy is average accuracy per class weighted by class size.

## 4. Results

### 4.1. CNN-2018

Overall validation accuracy of CNN-2018 is quite high—0.9050. The highest accuracy is reached for the ice free class (0.97) and the lowest for the young ice class (0.85) as shown in the confusion matrix in Figure 6.

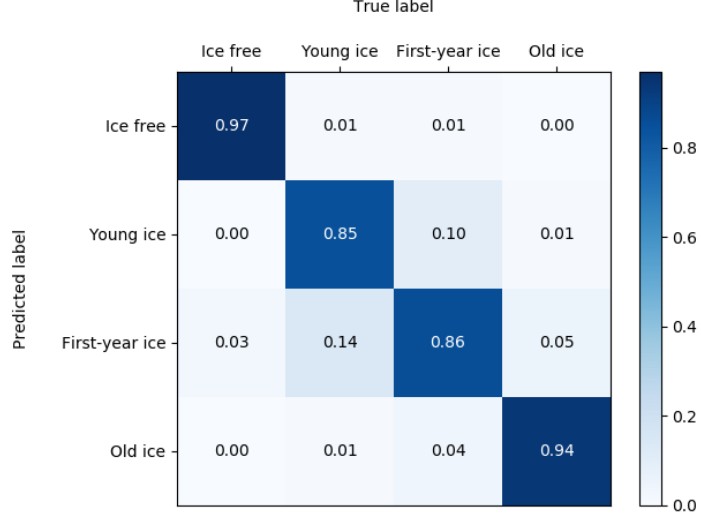

**Figure 6.** Confusion matrix for CNN-2018.

Comparison of manual ice charts with CNN-produced ice charts (see two examples on Figure 7) indicates that generally spatial distribution of ice types on both charts agrees very well. Obviously resolution of the CNN ice chart is much higher than the manual one and captures many small details including thin filaments of ice in open water, cracks in sea ice filled with young ice or water. The young ice category is more prevalent on the CNN ice charts than on the NIC ice charts. For example, it extends more along the ice edge on the first CNN ice chart (Figure 7C) and occupies a larger area on the second one (Figure 7H). Some water pixels are obviously misclassified as first-year ice in the waters roughened by strong winds (Figure 7H). New ice category present on NIC ice charts is mostly classified as open water by the CNN.

The classification probability maps (Figure 7E,J) show the probability of the selected ice type and can serve as a measure of CNN prediction uncertainty. The highest probabilities (0.9–1) are observed for the ice free and old ice classes. Young ice and first-year ice in general have lower probabilities (0.7–1). Probability of the misclassified first-year ice patch in the open water (Figure 7J) is relatively low (0.6–0.7) and may allow to discard incorrectly classified pixels.

The CNN method outperforms the ice type classification algorithm presented in [15] when applied to exactly the same data (see Table 3 for validation accuracy comparison). The combination of Haralick texture features and random forest classifier is a rather complex algorithm and takes more than one hour to process one Sentinel-1 SAR scene, whereas with the CNN it takes about 2 min.

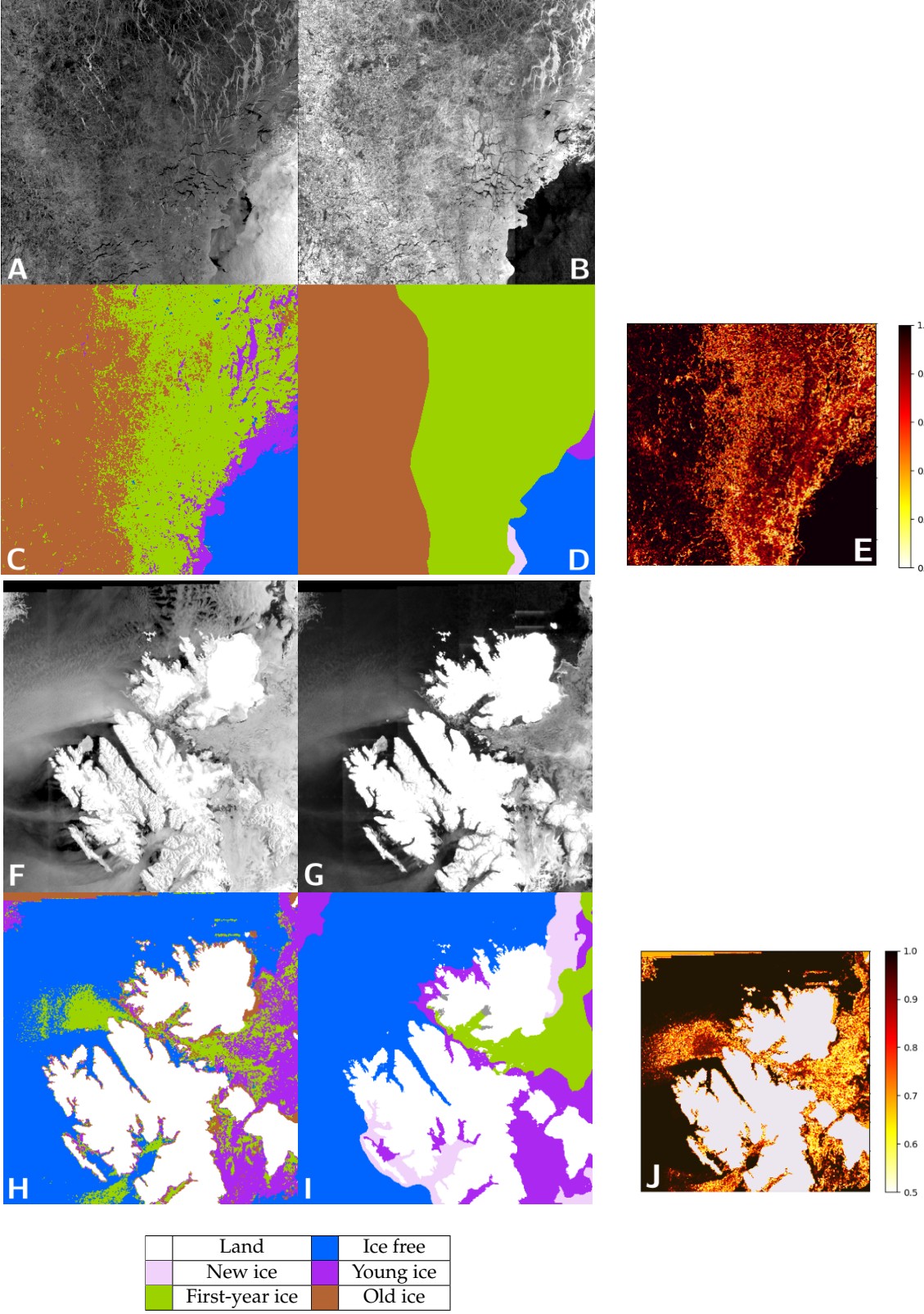

**Figure 7.** Comparison of CNN-produced and manual ice charts. (**A**,**B**,**F**,**G**) show Sentinel-1 SAR image in HH, HV polarizations acquired on 6-Mar-2018 at 07:43:19 and on 14-Feb-2018 at 06:22:54, correspondingly. (**C**,**D**,**H**,**I**) show CNN and manual ice charts for the corresponding dates. (**E**,**J**) show probability maps of the CNN-produced ice charts (values range from 0.5 to 1). Color-coding is provided in the table below images.

**Table 3.** Validation accuracy comparison between random forests and CNN.

| Accuracy Per Class (in %) | Ice Free | Young Ice | First-Year Ice | Old Ice |
|---|---|---|---|---|
| **Random forests** | 95.6 | 61.3 | 64.5 | 88.1 |
| **CNN** | 97 | 85 | 86 | 94 |

### 4.2. CNN-2020

Accuracy of CNN-2020 trained on 255 images is a bit higher—0.916 when computed over all samples in the validation dataset. The highest accuracy corresponds to ice free and old ice—0.98, the lowest to young ice—0.76.

If only samples with high enough probability are considered then accuracy increases even further. Figure 8 shows three confusion matrices computed for all samples (A), for samples with minimum probability of 0.5 (B) and 0.6 (C). Setting the probability threshold to 0.5 decreases the total number of valid samples only by 1.4 % and 0.6 by 4.6 %. Further increase of the probability threshold decreases the number of valid samples significantly and is not advised. Change in the probability threshold mostly affects accuracy of young ice and first-year ice predictions—accuracy grows from 0.76 and 0.84 to 0.79 and 0.89, correspondingly. Accuracy of ice free and old ice is almost not affected and stays rather high (0.98).

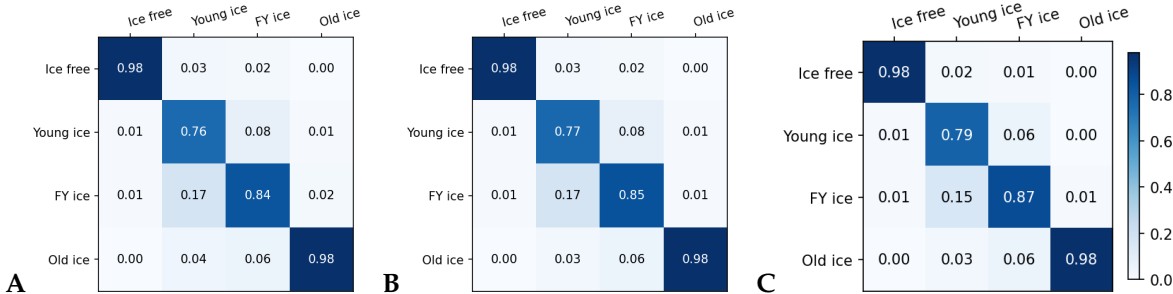

**Figure 8.** Confusion matrices for CNN-2020. (**A**)—all samples were used. (**B**)—only samples with maximum probability over 0.5 were used, (**C**)—samples with maximum probability over 0.6 were used. True labels are given on the X-axis, predicted labels on the Y-axis.

Individual SAR scenes were processed with CNN-2020 and the individual ice charts were reprojected on a grid in polar stereographic projection using Nansat [27] and aggregated into a single mosaic. Two mosaics generated from Sentinel-1 data for the 1st and 2nd April 2020 were compared with a NIC ice chart issued on 2nd April (Figure 9). The individual CNN ice charts were additionally processed with median filter with window size 5 × 5 pixels. Only pixels with probability over 0.5 were used.

Comparison of NIC and CNN ice charts again indicates very high level of general similarity. Moreover, the CNN predictions from 1st of April are quite similar to the 2nd of April showing high degree of CNN stability, notwithstanding significant increase in wind speed (from 5 to 16 m/s [28]) and brightness of ice free pixels, as can be seen on the second mosaic (Figure 9D). As previously mentioned, the CNN results have much higher resolution even after median filtering. Additionally, the CNN ice charts can be generated every day (NIC ice charts are provided only every week) and present important information about sea ice variability.

The few inconsistencies between NIC and CNN ice charts include the following. First-year ice is often predicted by CNN as young ice (e.g., south of Svalbard or along the ice edge in the Greenland sea). A strip of first-year medium ice on the first mosaic in the Greenland sea disappeared on the second mosaic. New ice is classified either as ice free or as young ice depending on actual ice conditions. Areas covered with fast ice were not included into training as they may contain many different ice types which do not drift. CNN predicts the presence of various ice types or even open water in these

regions. Although it is impossible to validate all pixels inside fast ice areas, the presence of open water near coasts is obviously wrong and can be explained by very smooth and dark-looking fast ice.

In contrast to the 2018-dataset, the model trained for the 2020-dataset is not tuned using the validation dataset. The validation dataset is used only to perform the early-stopping procedure (training stops after 15 epochs without improvement) aimed at reducing the computation time of the training. Experiments on the 2018-dataset showed that the early-stopping does not affect the final score (i.e., there is no over-fitting): the validation dataset is thus still valid to compute relevant classification scores.

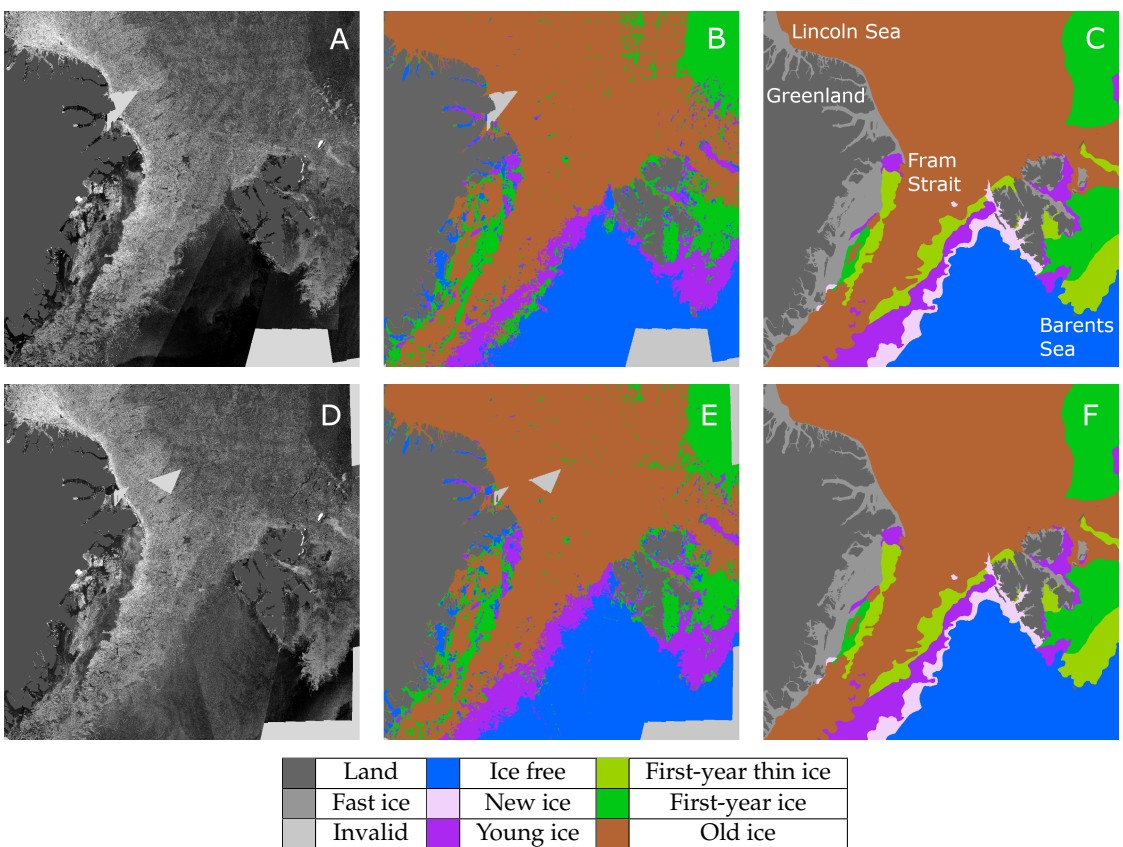

| | Land | | Ice free | | First-year thin ice |
|---|---|---|---|---|---|
| | Fast ice | | New ice | | First-year ice |
| | Invalid | | Young ice | | Old ice |

**Figure 9.** Mosaics of Sentinel-1 SAR HV images (**A**,**D**), CNN ice charts (**B**,**E**) and National Ice Center (NIC) ice chart (**C**,**F**) for 1-Apr-2020 (upper row) and 2-Apr-2020 (lower row). The new ice and first-year thin ice classes are present only on the NIC ice charts because these labels were not included in CNN classification dataset.

## 5. Sensitivity Study

This section addresses the sensitivity of the proposed approach to CNN hyperparameters (Section 5.1), preprocessing steps (Section 5.2) and thermal noise (Section 5.3).

### 5.1. CNN Hyperparameter Sensitivity

Table 4 presents the CNN hyperparameters considered in the sensitivity study. The set of all possible combinations of hyperparameters in Table 4 is denoted $\mathcal{H}$.

The number of possible hyperparameter combinations (the size of $\mathcal{H}$) in Table 4 is too large to be tested exhaustively. The exploration of the hyperparameters is thus performed using a so-called "randomized search" procedure [29] detailed in the Algorithm 1. In this algorithm, $N$ particular combinations of hyperparameters are drawn randomly from $\mathcal{H}$ (described in Table 4), and for each combination, a neural networks is trained. The other parameters used for the training are described in Table 5 and the architecture of the neural network is the one described in Figure 5. For each training,

the weights are recorded for each epoch, and the set of weights corresponding to the best accuracy on the validation set is selected. Note that in these CNN hyperparameter sensitivity experiments the sub-image size was equal to 25 pixels for increasing training speed. The sensitivity to sub-image size is studied in experiments shown in Section 5.2.

**Table 4.** Hyperparameters linked with the CNN.

| Name | Range Values Considered |
|---|---|
| Number of neurons per hidden dense layer. x is the number of the layer. | 16, 32, 64, 128, 256, 512, 1024 |
| Learning rate | 0.001, 0.01, 0.1 |
| Dropout rate $p_D$ | 0.1, 0.2, 0.3, 0.4, 0.5, 0.6, 0.7, 0.8, 0.9 |
| Convolutional size | 3,5 |
| Number of filters per convolutional layer. y is the number of the layer. | 32, 64, 128 |
| Batch size | 32, 128, 512, 1024 |
| $L_2$ regularization coefficient $\gamma$ | 0.001, 0.01, 0.1 |

---

**Algorithm 1:** Randomized search algorithm for hyperparameters.

---

**Input:** $\mathcal{H}$: Set of combination for hyperparameters,
　　　　$\mathcal{D}_{\text{train/val}}$: Training/Validation set,
　　　　$N$: Total number of experiments
**Output:** $N$ trained networks.
**Repeat $N$ times**
　　Draw a set of hyperparameters $h$ from $\mathcal{H}$;
　　Initialize the neural network with $h$;
　　Train the neural network using $\mathcal{D}_{\text{train}}$;
　　Compute the score using $\mathcal{D}_{\text{val}}$;
　　Store the results;
**end**

---

**Table 5.** Parameters of experiments.

| Data Hyperparameters | Training Parameters |
|---|---|
| sub-image size: 25 pixels | number of epoch: 50 |
| number of classes: 4 | number of training samples: 210,000 |
| boundary distance: 20 pixels | number of validation samples: 90,000 |

After performing the $N = 50$ trainings, we can finally assess the sensitivity to one particular hyperparameter mentioned in Table 4. For example, to assess the effect of the regularization parameter $\gamma$, the 50 trained networks are split into 3 subsets, each one with a unique value of $\gamma$: 0.001 for the first subset, 0.01 for the second and 0.1 for the third one. The accuracy is then computed in each subset on the validation set. The results for the most sensitive hyperparameters are presented in Figure 10. The most sensitive parameters are the learning rate, the dropout rate and the batch size. Obviously, our choice of hyperparameters described in Section 3 corresponds to the best set of hyperparameters. Interestingly, the best dropout rate is the smallest value, 0.1, which has the lowest regularization effect. It could suggest that our problem is not highly impacted by overfitting.

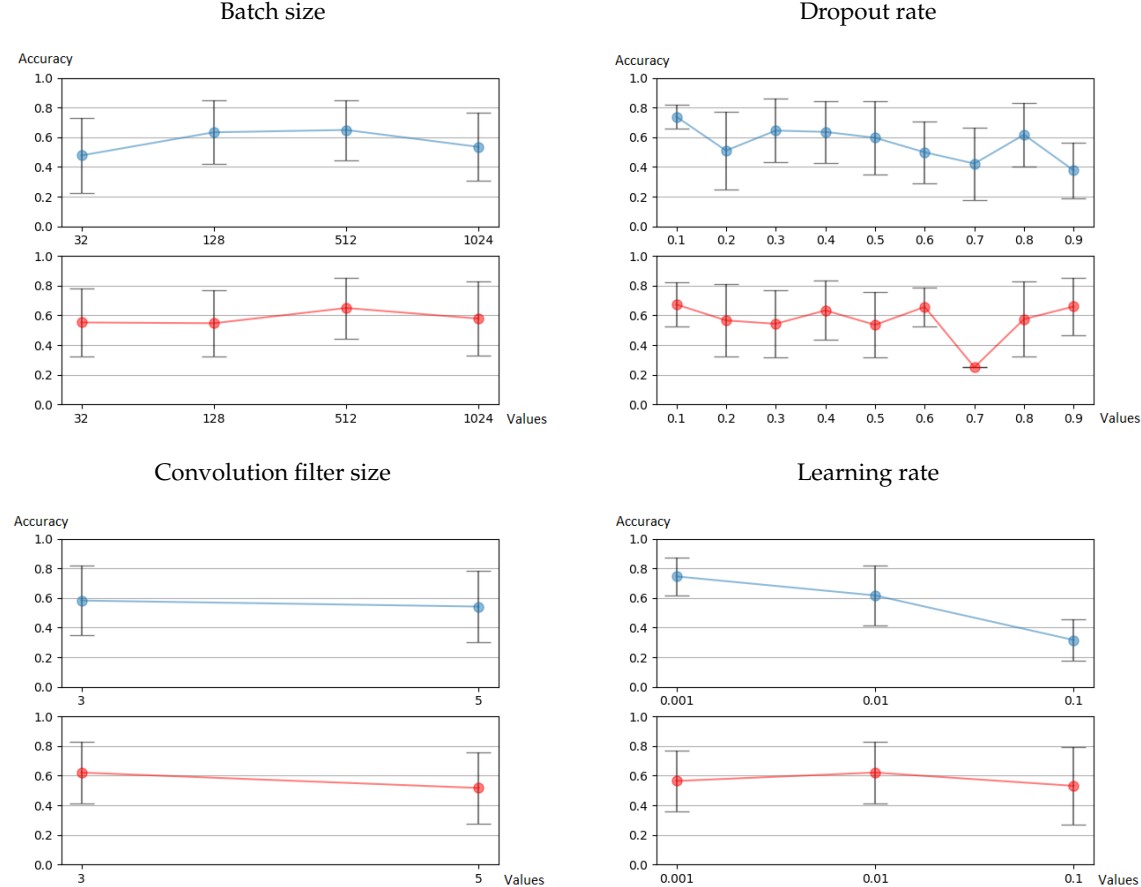

**Figure 10.** Dependence of the CNN average validation accuracy on batch size (**top left**), convolution filter size (**bottom left**), dropout (**top right**) and learning rate (**bottom right**) for architectures A (blue) and B (red). The vertical bars correspond to $\pm 1$ the standard deviation computed on the subset of the 50 training results.

The sensitivity to the number of filters of the convolutional layers and to the regularization parameter $\gamma$ is not shown but is not very strong. The sensitivity is also very low to the size of the convolution (lower left panel in Figure 10). The parameters selected for the final training described in Section 3 are the less computationally costly, i.e., a convolutional size of 3, 32 convolutional filters and the size of the dense layer as indicated in Figure 5. There is also a very weak sensitivity to the number of neurons on the 3 dense layers (not shown). The architecture selected is the one corresponding to the best combination of these hyperparameters (see Figure 5).

A sensitivity experiment to the architecture of the neural network itself was also conducted. Another architecture with the same number of layers and with a different sequence of convolutional and pooling layers was tested. The resulting accuracy was similar to the original sequence, the training was longer by a factor 1.6. Another training was also performed by removing one convolutional layer, or by removing dense layers, but the accuracy was impaired (not shown here).

*5.2. Sensitivity to Preprocessing Parameters*

In this section the effects of two critical parameters on the preprocessing summarized in Section 2.3 is addressed. The first parameter is the size of the sub-images considered in the input of the neural network. A big size gives all the necessary information to classify but can add noise if a sub-image contains a mixture of classes. A small size of the sub-image gives more spatially accurate information but can fail to provide enough spatial context to the ice-chart pixel. The second parameter is the distance used to withdraw from the training set pixels that are too close to the border to another class.

If this distance is too small, sub-images in the dataset can be misclassified or have mixed ice type within the sub-image. If this distance is too large, too many samples from the training set could be withdrawn. Table 6 summarizes the range of values tested for these two parameters. A full training was performed for each possible combination of these parameters on 20 epochs with 140,000 samples for training and 60,000 for the validation (which is sometimes referred to as a grid-search procedure). The same SAR images were used to produce the validation dataset for each combination of the preprocessing parameters.

The performance is summarized in Table 7. Spatial information is important and a sub-image of 50 × 50 provides better results. In addition, in our case, a correct cleaning of the training set by avoiding sub-images with a mixed type of ice has a significant effect to improve the performance of the training. Additionally noteworthy, the quality of the training set has a greater impact on the performance than the choice made for the construction and the training of the neural network.

**Table 6.** Preprocesing parameters.

| Name | Description | Value |
| --- | --- | --- |
| **Distance to class border (in pixels of the ice chart image)** | All the pixels of the ice chart under this distance are deleted to avoid class mixing. | 5, 10, 15, 20 |
| **Sub-image size (in pixels of the SAR image)** | The size of the SAR sub-image which is linked with one pixel from ice chart. | 25, 50 |

As mentioned in Section 2, one ice chart pixel has to match with one SAR sub-image. In consequence the pixel's resolution of ice chart image is 1 km for 25 × 25 SAR sub-images and is 2 km for 50 × 50 SAR sub-images. Thus a fair comparison of the class distance parameter has to be done in kilometers. Table 7 shows accuracy for different values of distance to the class border as well as SAR sub-image size: for the same kilometer distance, 50 × 50 images give the best results and for both image size, accuracy is better when border distance increases. Note the improvement of accuracy with the distance to the class border is due to two factors: (i) the training set includes less mixing pixels and so the training is not impaired by noisy labels; (ii) the validation dataset includes also less noisy labels, and so the performance is evaluated more robustly.

**Table 7.** Sensitivity of accuracy with the preprocessing parameters.

| Distance to Border (Pixel) | Real Distance (km) | Accuracy |
| --- | --- | --- |
| **Sub-image size : 25 × 25** | | |
| 5 | 5 | 0.7392 |
| 10 | 10 | 0.7583 |
| 15 | 15 | 0.7717 |
| 20 | 20 | 0.7843 |
| **Sub-image size : 50 × 50** | | |
| 5 | 10 | 0.7983 |
| 10 | 20 | 0.8278 |
| 15 | 30 | 0.8636 |
| 20 | 40 | 0.8966 |

*5.3. Impact of Thermal Noise and other Properties of Input Data*

Removal of thermal noise from SAR data [15–17] is a costly task and the sensitivity of our method to the noise in the SAR images was addressed. A CNN with the same architecture as on Figure 5 was trained on raw SAR images (with no denoising performed). The final accuracy is 0.89, which is close to

the accuracy obtained with denoised image 0.92. The denoising allowed improvements mostly for the ice-free class, see for example, Figure 11. It shows that, even if denoising can improve the final accuracy, the selected neural network has the potential to process noisy satellite images which offers promising perspectives for the segmentation of SAR images.

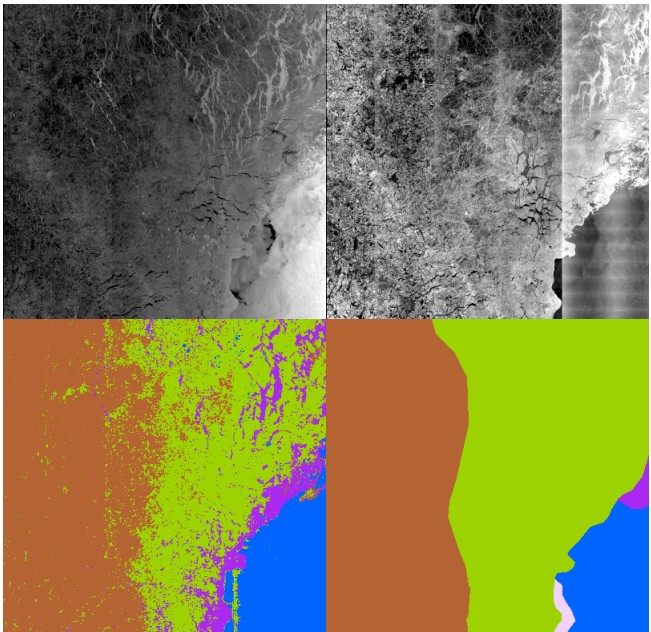

**Figure 11.** A comparison of manual and CNN-produced ice charts with noisy input data. Two top images show Sentinel-1 SAR image (6-Mar-2018 at 07:43:19) in HH (**left**) and HV (**right**) polarizations. **Lower left**—CNN predicted ice chart with noisy inputs in the same projection as SAR image, **lower right**—manual ice chart. See Figure 7 to compare with the processing using denoised data. Accuracy of the CNN-produced ice charts is 0.79 and 0.80 when compared to the NIC ice charts.

As can be seen from comparing the overall accuracy of CNN-2018 and CNN-2020 the impact of satellite data precision and texture noise removal algorithm is not significant. The slightly different preprocessing steps for the 2020-datasets were dictated by practical considerations: simplification of processing chain and better visual appearance of SAR images.

## 6. Discussion

The observed mismatch between ice type labels from the ice charts and CNN predictions can be partly explained by low resolution of NIC ice charts and inaccuracies in manual classification. Obviously, it is impossible for an ice expert to draw an absolutely accurate ice chart depicting all elements. Moreover, ice charts sometimes tend to overestimate severity of sea ice conditions by increasing the age of the ice or extent of ice cover for assuring safer navigation. An example of manual and CNN ice classification on Figure 12 shows that the new ice category mostly covers open waters and in some places young ice extends into open water by 5–15 km. In addition, the boundary between young-ice and multi-year ice is drawn with limited precision and both classes are mixed near the boundary. Inconsistencies between ice charts were also observed in previous studies (see, for example, Figure 9 in [15]). The largest and most obvious of these inconsistencies can be corrected manually prior to CNN training but a lot will still remain in the training dataset.

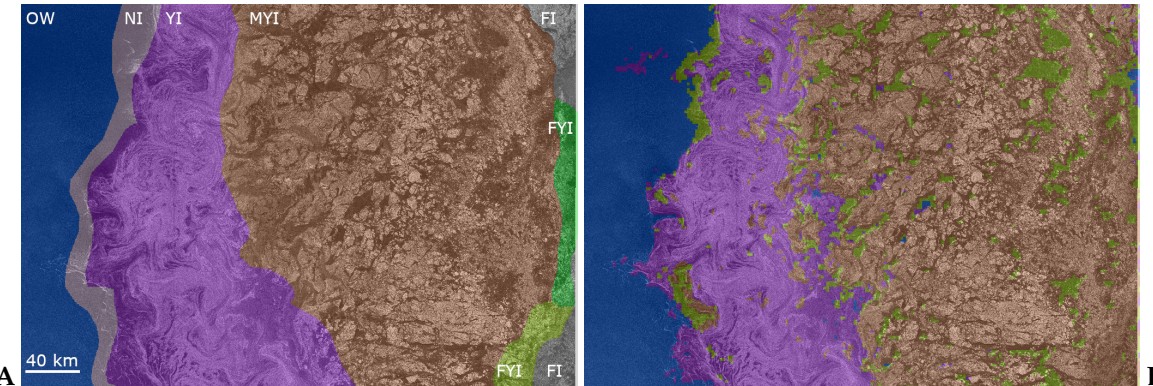

**Figure 12.** NIC (**A**) and CNN (**B**) semi-transparent ice charts on top of Sentinel-1A SAR image in HV polarization taken on 2020-01-01 07:46:20. Labels on the NIC ice chart denote the following types: OW—open water, NI—new ice, YI—young ice, FYI—first-year ice, MYI—multi-year ice, FI—fast ice. Colors on the NIC and CNN ice charts correspond.

The CNN approach to sea ice type classification is split in two phases: (i) semi-automated training phase and (ii) fully automated application phase. The first phase consists of two major steps—data preparation and CNN training. Data preparation is a tedious and time consuming process that must include human judgment both for labeling data and for quality control. In our study we have used the NIC ice charts and overcame the first bottleneck—at least on our side no extra effort is needed for the labeling procedure. Quality control of the training data is also automated as much as possible using the "Distance to border" and "CNN segment size" filters (see Section 2.2). However, the initial selection of SAR scenes that match the best with the NIC ice charts still remains to be done manually and is the main hurdle in updating the automated algorithm.

In the second phase no human involvement is expected: SAR data are first pre-processed with calibration and noise removal and then CNN is applied in just few minutes (see Section 4.1).

Theoretically, there is a need to train a CNN only once and then it should be applicable to any input SAR data of the same type. In practice, however, a universal CNN is not possible and several new CNNs may need to be trained. This may happen, for example, in the case when preprocessing of the input SAR data has to be changed due to update of the software on a ground receiving station and changes in the signal or the annotated thermal noise radiometric accuracy. As pointed out in [16], ESA has been changing the calibration of the thermal noise vectors in Sentinel-1 SAR several times in recent years and if that continues to happen in future the CNN will have to be retrained. Another challenge with a universal CNN is seasonal cycle of SAR backscatter signature. In spring, when melting starts in the Arctic, the changes in porosity, density and wetness of snow cover and appearance of melt ponds affect the total power of SAR signal and texture [30] which may require an extra CNN for summer season to be trained.

Despite the aforementioned challenges, the operational CNN ice type product opens up interesting possibilities. First, of course, is the utilization of automated ice charts for safe navigation in ice infested waters. Compared to the traditional manual ice charts, the CNN charts have higher spatial resolution and most importantly higher frequency. Near real time information about ice conditions is crucial both for tactical route planning for ice breakers or ice strengthened ships and also for long term route planning for ordinary open-water ships. In this context automatic separation of several ice types is especially important for identification of navigable waters according to the Polar Code [31].

The second potential application, assimilation of sea ice type information into numerical models is also of high relevance. In this regard, a correct estimation of sea ice types uncertainty is necessary to be used efficiently in e.g., ensemble Kalman filter [32]. As a first stage, it would be possible to consider the output probability as uncertainty estimate as it covers major uncertainties and misclassification cases (see Figure 7). Nevertheless, it was shown, e.g., in [33], that using the raw output probability can lead to underestimating the uncertainty. Methods were proposed [34] to better quantify this uncertainty,

involving very few modifications to the training algorithm, and may have to be implemented in a later stage. Information about three ice types is also essential to constraint numerical models. Sea ice in these categories obviously has different thickness, roughness, mechanical and thermodynamic properties. With proper parametrization in sea ice models, assimilation of ice type information should influence the drag coefficients, cohesion, elasticity, viscosity, thermal conductivity and, consequently, forecast of ice drift, concentration and thickness.

In addition to reasoning above there is one more consideration why detection of three ice classes may have benefit over only two classes. In the experiments (not shown here) with combination of different ice classes we noticed that sometimes when young-ice and first-year ice classes are combined, the accuracy does not increase or even decreases. A potential reason is that in the latent space of CNN internal data representation these classes are not close to each other or, maybe, closer to open water or multi-year ice. Hence, combination of young-ice and first-year ice in training data leads to reduction in quality of water classification. Nevertheless, further investigations on these topics require much more effort and are outside the scope of the current study.

## 7. Conclusions

Our study demonstrates that CNN can be successfully applied for classification of sea ice types in SAR data. A CNN-based algorithm was developed and validated on data from 2018 (same as in [15]) and 2020. The algorithm is applied in small sub-images extracted from a SAR image after preprocessing including thermal noise removal. Validation shows that the overall accuracy exceeds 90% and errors are mostly attributed to coarse resolution of ice charts or misclassification of training data by human experts.

Several sensitivity experiments were conducted for testing the impact of CNN architecture, hyperparameters, training parameters and data preprocessing on accuracy. It was shown that a CNN with three convolutional layers, two max-pool layers and three hidden dense layers can be applied to a sub-image with size $50 \times 50$ pixels for achieving the best results. It was also shown that a CNN can be applied to SAR data without thermal noise removal on the preprocessing step. Understandably, the classification accuracy decreases to 89% but remains reasonable.

The main advantages of the new algorithm are the ability to classify several ice types, higher classification accuracy for each ice type and higher speed of processing than in the previous studies. The relative simplicity of the algorithm (both texture analysis and classification are performed by CNN) is also a benefit. In addition to providing ice type labels, the algorithm also derives the probability of belonging to a class. Uncertainty of the method can be derived from these probabilities and used in the assimilation of ice type in numerical models.

Given the high accuracy and processing speed, the CNN-based algorithm will be proposed to the Copernicus Marine Environment Monitoring Service (CMEMS) for operational sea ice type retrieval for generating ice charts in the Arctic Ocean. It is already released as an open source software [26] and available on Github: https://github.com/nansencenter/s1_icetype_cnn.

**Author Contributions:** Conceptualization, H.B., A.K. and J.B.; methodology, H.B., A.K. and J.B.; software, H.B. and A.K.; validation, H.B., A.K. and J.B.; formal analysis, H.B. and A.K.; resources, A.K.; data curation, H.B. and A.K.; writing—original draft preparation, H.B., A.K. and J.B.; visualization, H.B. and A.K; supervision, A.K. and J.B.; project administration, A.K.; funding acquisition, A.K. All authors have read and agreed to the published version of the manuscript.

**Funding:** This work was supported by the French Service Hydrographique et Océanographique de la Marine (SHOM) under SHOMImpSIM Project, 111222.

**Acknowledgments:** We thank Anastase Charantonis (École Nationale Supérieure d'Informatique pour l'Industrie et l'Entreprise, France) for his helpful advice and comments during the planning of the CNN hyperparameter experiments.

**Conflicts of Interest:** The authors declare no conflict of interest. The funders had no role in the design of the study; in the collection, analyses, or interpretation of data; in the writing of the manuscript, or in the decision to publish the results.

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
