# Peer review of "Classification of Sea Ice Types in Sentinel-1 SAR Data Using Convolutional Neural Networks"

_remotesensing, doi:10.3390/rs12132165_

Round 1

Reviewer 1 Report

Specific comments are provided in the attached pdf with highlights and comment notes.

Overall comments:

Generally, I think the paper presents a novel contribution to the classification of sea ice types that have several advantages over previous methods presented for this. Given that comments are addressed I think it will be of interest to the sea ice community.
Due to the number of comments I have and the severity of some of them, I have chosen to mark this as a major revision, but I strongly encourage the authors to resubmit with the requested changes.

For the readability, the paper could benefit from a language-edit, but it is mostly regarding the style and does not hamper the understanding of the content.

The section describing the dataset is confusing and unclear. It is not clear on which criteria pixels marked by experts are found invalid. It is also confusing that two different ways of storing the Sentinel-1 data are done for 2018 and 2020 datasets, e.g. different precisions, noise-removal/no noise-removal, etc.
A natural question is, what is the influence of the numerical precision on performance?

It is very hard for me to understand the part about "CNN segment size" and the temporary network used is not explained at all. I recommend you revise this paragraph (125-131).

I am worried about some quite crucial points regarding the independence of the validation data, and one of the concerns is whether the random split is done properly. If there is any overlap between samples taken out, a random split is not a valid method to ensure independence.

The other concern is regarding the criteria for stopping the CNN training. If you measure loss/acc on the validation data, then this data is no longer completely independent and you split and have a 3rd "test-set" which only is used to test after training is finished.

Reviewer 2 Report

This manuscript has provided a novel and practicable approach to classify sea ice types by applying the CNN algorithm to Sentinel-1 data. The text in this paper is well written, the figures/tables are neatly presented, and the content is clearly organized. The related works and studies (reference) regarding to this topic are appropriately cited. The experimental methodology is quite understandable, the results are convincing and well discussed. The conclusion provides good advices for the current and future works. The corresponding codes are open to the public. I recommend publishing this paper after minor revision.

Comments in detail:

Lines 14-15: it's a misleading definition of “Ice concentration”, suggest some thing like "the ratio of sea ice pixel to total pixel area of water"?

Lines 46-50: It’s suggested to add a figure of geographic map showing (1) study area; (2) overlaid data range, i.e. the 44 SAR scenes (frame only, not image).

Line 63: provide a couple sentences to briefly describe the algorithm used in [15], emphasizing any difference to the method used in this manuscript.

Line 90: What’s the scale factor of the ice charts, i.e. the approximate spatial resolution of the original ice chart?

Figure 2 (left): the effect of the radio frequency interference is hard to see.

Line 206: please explain the meaning of the accuracy value [0,1]?

Line 220: “(B in figs 6 and 7)”, but there are no (a) and (b) indicated in Figures 6 and 7.

Figure 7 caption: better to use “Same as in Figure 6, but for 15-Feb-2018 at 06:22:54.”

Figure 8: use labels (a) (b) and (c) instead of Left, Middle and Right.

Line 268: use “This section” instead of “The following section”

Line 282: Is it a figure or table? Perhaps reform it to a flow chart figure.

Figure 10: use labels (a) (b) (c) and (d) to indicate the four diagrams.

Reviewer 3 Report

This paper describes a method for classifying pixels in Sentinel-1 Extra Wide Swath SAR images of the Arctic Ocean using a convolutional neural network (CNN). The classes are: ice-free, young ice, first-year ice, and old ice. The study region stretches from the Canadian Arctic Archipelago to Franz Joseph Land between latitudes 75N and 85N. There are two study periods: winter 2018 and winter 2020. The CNN is trained on digitized ice charts from the U.S. National Ice Center. The ice charts are digitized with a grid cell size of 1 km or 2 km. The SAR images are 400 x 400 km with pixel size 40 m (thus 25 x 25 or 50 x 50 SAR pixels per ice-chart cell). The paper describes the data processing and the CNN architecture. Results are given in Figure 5 for 2018 data and in Figure 8 for 2020 data. The classification accuracy is high, and the method outperforms a previously published algorithm. The paper examines the sensitivity of the CNN to several of its parameters.

In general, this paper is a worthwhile contribution to sea-ice classification and the use of neural networks, and it should eventually be published. However, I feel that there are several missing pieces and many confusing details that need to be fixed before it is ready for publication, i.e. it needs major revision.

I should say that I am not an expert on neural networks. My comments on the CNN may need to be viewed in that light.

General Comments

In spite of the fact that Section 5 is labelled "Discussion", it is not a discussion section. It is about the sensitivity of the CNN to several of its parameters. A proper discussion section is missing, and is definitely needed. Among the issues that need to be discussed are: (1) What are the prospects of applying this method routinely to SAR images? That is, what hurdles (if any) still need to be overcome? The authors note that they performed manual corrections to some SAR images before applying the CNN, and that this was "very time consuming" (line 113). How is it possible, then, to propose that this algorithm be used for "operational sea ice type retrieval" (line 360) when time-consuming pre-processing is necessary? (2) The manual corrections are made to the Extra Wide Swath SAR images at the internal boundaries where different beams are stitched together. If the CNN method were applied to ordinary (not Extra Wide) SAR images, presumably the manual corrections would not be necessary. Is this feasible? What are the pros and cons? (3) It seems that if the young ice and first-year ice classes were combined into one class, the accuracy of the three remaining classes would be well over 90%. Is that correct? Is that a reasonable alternative to the current four classes, or is it critical to distinguish young ice from first-year ice? See also the next comment.

The Introduction needs to address the question of why it's important to distinguish the four classes of pixels -- in particular, why young ice needs to be distinguished from first-year ice. Clearly it's important to distinguish ice-free from ice-covered pixels, and it's well known that old (multiyear) ice is generally thicker and harder than first-year ice, so that distinction is important for navigation, but what is the importance of distinguishing young ice from first-year ice? Is it just a matter of ice thickness?

The Introduction should also say why a simple thresholding of the SAR backscatter cannot adequately separate the desired classes. In calibrated SAR images, old (multiyear) ice has a very different signature from open water, so those two classes could be separated simply by applying a threshold to the calibrated backscatter. Is it not possible to distinguish a third class between ice-free and old ice by simply applying thresholds? Is that what drives the need for a sophisticated approach such as CNN vs. a simple threshold? On a related note, I see that the CNN method failed to correctly classify open water that had been roughened by the wind (line 218 and Figure 7), so CNN made the same mistake that a simple thresholding approach would make.

Specific Comments

Lines 14-15. "Ice concentration is defined as the ratio of total pixel area to the area covered by sea floating ice." This is exactly backwards. Ice concentration is the ratio of the area covered by sea ice to the total area.

Lines 46-48. A map would be helpful, though not essential.

Equation (2). I think the left-hand side should be w2, not w1.

Lines 121-124.
(a) I don't understand the difference between the i and k subscripts. Why is it "y_i" on line 122 but "y_k" on line 123? (b) The vector y_i is supposed to be of length C=4, but 5 components are given.

Line 134. "These vectors are used to select only valid data from the input and output data sets." The vectors do not SELECT valid data, they DEFINE valid data.

Lines 150-151. There are 5 un-numbered lines between 150 and 151. About these 5 lines: they are too sketchy (not enough detail). What is the variable x? What is its range? Why is the "max" function used?

Equation (4). According to line 169, the p_k are all between 0 and 1, and their sum is equal to one. With C=4, the denominator is
e^1 + e^2 + e^3 + e^4. The p_k can only add up to 1 if the vector z is a permutation of the numbers 1,2,3,4. It's not clear to me that this is the case. Why is the vector z a permutation of 1,2,3,4?

Section 3.2. In this section, the names and corresponding symbols of the parameters are not clearly spelled out.
(a) In equation (5), what is gamma?
(b) On line 182, what is the symbol for the learning rate?
(c) On line 183, the "L2-regularization" is gamma. But line 176 refers to "L2-regularization term weighted by gamma". Does "L2-regularization" refer to a parameter or to a term in equation (5)?
(d) In Table 1, give the symbol of each parameter.

Line 177. N_theta = 72,008. How does this follow from the CNN architecture (Figure 4)?

Lines 195-196. "the following metrics on the validation dataset are computed (as described in section 2.3)" Section 2.3 does not describe computing anything. Perhaps delete "(as described in section 2.3)".

Lines 197-203. There are three metrics: overall accuracy, class accuracy, and confusion matrix. From the confusion matrix, can we calculate the overall accuracy and the class accuracy? If yes, how? Are the class accuracies given by the numbers on the diagonal of the confusion matrix? Does each column of the confusion matrix add up to 1?

Line 216. "Young Ice category is more present on ice charts." This sentence is ambiguous because both the NIC ice charts and the CNN-produced classified images are called "ice charts". See line 212, and the first line of the caption of Figure 6. Maybe the sentence on line 216 should read: "The Young Ice category is more prevalent on the CNN ice charts than on the NIC ice charts."

Lines 247-248. "The individual scenes were additionally processed with median filter with window size 5 x 5 pixels." Please clarify whether this applies to the SAR images or the CNN-produced ice charts or the NIC ice charts.

Lines 252-253. "notwithstanding significant increase in wind speed and brightness of ice free pixels, as can be seen IN THE LOWER PART OF the second mosaic (FIGURE 9, PANEL X)." What is the wind speed in this case where the CNN did correctly classified the pixels as open water, vs. the wind speed in Figure 7 where the CNN did not correctly classify the pixels due to wind-roughening?

Figure 9. Here we have "First-year medium ice" in the CNN ice charts, which is NOT the name of one of the four ice classes. What is the difference between "First-year ice" and "First-year medium ice"? The caption says, "The New ice and First-year ice classes are present only on the NIC ice charts because these labels were not included in CNN classification dataset" -- this is not correct; "First-year ice" is most definitely one of the labels included in the CNN classification. Please clarify the terminology here.

Line 267. Section 5, Discussion. As noted above in the General Comments, this is the sensitivity section, not the discussion section. Please come up with a proper title for this section.

Table 4. The Acronym column is very confusing.
(a) "Nx" is not used anywhere in this paper.
(b) "LR" is not used anywhere in this paper.
(c) "D" is not used for the dropout rate. "p_D" is used.
(d) "CS" is not used anywhere in this paper.
(e) "Cy" is not used anywhere in this paper.
(f) "B" is not used for batch size. "n" is used.
(g) "L2" apparently stands for "L2 rate" (in the Name column) which is not very helpful. Perhaps "L2 rate" is supposed to be "L2-regularization rate"?
(h) The symbol gamma is used in the table for "Learning rate". But on line 183 gamma is "L2-regularization"; on lines 284-285 gamma is "regularization parameter"; and on lines 292-293 gamma is "regularization rate".

Line 280. Is one "epoch" the same as one run of the CNN?

Table 5. Is "number of epoch" the same as "N" on line 283?

Line 286. Here it says that the value of gamma for the second subset is 0.02, but in Table 4 the value is 0.01.

Figure 10. (a) The four sets of plots should be labelled (Batch Size, Dropout Rate, etc). (b) The caption says that "convolution filter size" is "top right" but it should be "bottom left". (c) The caption says that "dropout" is "bottom left" but it should be "top right". (d) The caption needs to explain the vertical bars in every panel. Do they represent +/- 1 standard deviation? Computed from 50 runs?

Lines 317-318. "it is relevant to choose a sub-image of 50 x 50." I don't understand what is meant by "relevant" in this context.

Line 345-346. "errors are mostly attributed to coarse resolution of ice charts or misclassification of training data by human experts." But see line 112: "pixels with obviously wrong classification by ice expert are also blanked out (see fig. 3)." So if the incorrectly classified pixels are removed beforehand, then how can errors in CNN be attributed to them?

Lines 357-358. "These probabilities can be interpreted as uncertainty of the method and used in assimilation of ice type in numerical models." Are ice types currently being assimilated in some numerical models of sea ice? If yes, please give a reference. If no, perhaps this should go in the discussion section with a few more details.

Technical Corrections and Comments

Title. Better to write "IN Sentinel-1 SAR data" than "ON Sentinel-1 SAR data"

Line 7. Change "of accuracy" to simply "accuracy" (two times)

Line 47. Change "Linkoln" to "Lincoln"

Line 65. Change "in details" to "in detail"

Line 75. Perhaps "corrected TO thermal noise" should be "corrected FOR thermal noise"?

Line 80. Change "fall OF the scope of this paper" to "fall OUTSIDE the scope of this paper"

Line 98. The WMO codes for first-year ice (87-94) do not match the table below Figure 1, in which the code for first-year ice is 86.

Figure 1. In the table below the figure, it's hard to read "Ice-free" on the dark blue background, and "Young ice" on the dark purple background, and "Old ice" on the brown background. Perhaps use lighter colors, or white font, or put the labels next to the color bars instead of inside the color bars.

Line 118. Change "IN the order of" to "ON the order of"

Line 145. Change "The figure 4" to "Figure 4"

Line 150. Delete "the"

Lines 150-151. There are 5 un-numbered lines between 150 and 151.

Line 154. Change "layers are performing" to "layers perform"

Lines 166-167. There are 4 un-numbered lines between 166 and 167.

Line above equation (4). Change "function, it computed the output" to "function. It computes the output"

Lines 172-173. There are 3 un-numbered lines between 172 and 173.

Line 183. Change "The table 1 is giving" to "Table 1 gives"

Line 208. Change "on" to "in". Usually "in Fig X" sounds better.

Line 220. "B on figs. 6 and 7" -- there is no "B" on the figs. However, it would be a good idea to label all the panels with letters (a), (b), etc.

Line 222. End the sentence after the word "classes." (period) and start a new sentence with "Young ice..."

Line 223. "in the open water (FIG 7) is relatively low..."

Figure 6. Same comment as for Figure 1 above.

Figure 6 caption, 3rd line. "lower right" should be "lower middle"

Figure 6 caption, 4th line. "the probability map OF THE CNN-PRODUCED ICE CHART"

Line 230. "hour TO process one Sentinel-1 SAR scene."

Line 235. Change "concerned" to "considered"

Lines 236-237. The text refers to panels A, B, C, of Figure 8, but the panels in the figure are not labelled. They should be.

Line 239. Change "is not applicable" to "is not advised"

Line 257. Change "Few inconsistencies" to "The few inconsistencies"

Line 261. Change "as it may contain" to "as they may contain"

Line 262. "CNN predicts THE presence of..."

Line 263. Change "but" to "the"

Figure 9. Same comment as Figure 1 above.

Lines 268-270. It's not necessary to put the word "the" before "section X". Also, check with the journal to see whether or not the words "section" and "figure" and "table" should be capitalized when they refer to specific sections and figures and tables in the paper.

Line 294. Change "down-left" to either "lower left" or "bottom left"

Line 300. Replace the comma (,) at the end of the line with the word "and"

Line 301. Change "accuracy is similar" to "accuracy was similar"

Line 313. Change "The table 6 summarized" to "Table 6 summarizes"

Line 320. Change "Noteworthy," to "Also noteworthy,"

Table 6, Description, upper box, change "All the pixel" to "All the pixels" (plural)

Line 332. "Removal OF thermal noise"

Line 336. "see for example an example of processing in Fig. 11." Maybe change this to something like "see an example of processing in Fig. 11." or "see for example the processing in Fig. 11." or "see for example Fig. 11."

Line 337. Change "the neural networks selected has the" to either "the neural networks selected have the" (i.e. plural) or "the neural network selected has the" (i.e. singular).

Line 337. Change "potentiality" to "potential"

Line 338. Change "image" to "images"

Line 342. Change "on SAR data" to "in SAR data"

Line 343. Change "extracted from SAR image after a" to "extracted from a SAR image after"

Line 347. "for testing THE impact of"

Line 352. Change "Understandingly" to "Understandably"

Line 361. Capitalize "Arctic Ocean"

Line 370. Change "advises" to "advice"

Line 406. Please check whether this "Discussions" paper (Park et al. 2019) has been accepted yet.

Line 418. The "Seger" reference is incomplete. Please provide the name of the journal, etc.

Round 2

Reviewer 1 Report

The paper has been largely improved and generally most comments have been addressed sufficiently. I do have some concerns still regards methodology and description thereof.

  1. You claim that it is fine not to do a three-fold split of your dataset, despite general consensus in the field of ML that this is necessary when you use the validation set for e.g. early stopping or hyper-parameters opt. Please refer to the reason for doing this is to compare with (Park et al.) but that generally testing could be "more" independent if a 3-fold split had been done.
    I believe in your results so I don't want to force you to retrain everything, but it is important to tell the field what is best practices!
  2. Please provide the stopping criteria for the early stopping scheme.
  3. It would be a nice addition to the paper if you calculated the accuracies in percent over the mosaics (figure 9) and provided these in e.g. the figure caption.
  4. In conclusion you are talking about using the probability output from the model as uncertainties in assimilation. You are "on thin ice" ;) here because NN probabilities are generally very over-determined due to the fact that we train on binary labels 0/1 for each class. There are a large amount of literature on uncertainties in deep learning which all agree that the output probability is not a good one. Please do not conclude that these probabilities can be used as uncertainties without backing it up by experiments, distribution plot, or references to other literature that had success with this.
    Instead, I suggest you refer to the literature of other methods on this. (e.g. there are a lot of work on this by the young assistant professor at oxford called Yarin Gal).

Reviewer 3 Report

Reviewer #3

The authors have addressed all my comments. In particular, the new Discussion section is very helpful. Below are a few minor technical corrections. After these have been made, I recommend publication. I do not need to see an updated version of the manuscript.

Minor Technical Corrections

Lines 18-20. "Ice type determines several ice characteristics including ice thickness, surface roughness, mechanical properties and is..." Ice type does not *determine* ice thickness, surface roughness, or mechanical properties. Ice type is simply a label that reflects some of those properties. Consider writing "Ice type is linked to several ice characteristics..."

Line 51. "Figure 9" -- change to "Figure 9C" or "Figure 9,C"

Figure 9C. The label "Barents Sea" is in the wrong place -- that's the Greenland Sea. Also the font size is too small.

Line 103. "As the results," -- change to "As a result,"

Line 154. Delete "only"

Line 173. "is evaluated" -- change to "are evaluated"

Line 180. "two trainable parameters" (plural)

Line 180. "It has two..."
If "It" refers to "Batch-norm layers" then write:
"They have two trainable parameters... They are used to facilitate the training, and also have a..."
If "It" refers to "the output of the previous layer" then write:
"It has two trainable parameters... It is used to facilitate the training, and also has a..."

Line 192. "the parameters" (plural)

Lines 217-218. "The two most sensitive parameters of the optimizers are the batch size n and the learning rate." I see the batch size n in equation (5). Where is the learning rate?

Figure 7 is missing from the revised manuscript. Apparently the old Figures 6 and 7 have been combined into one new figure with panels A through J. That's fine, and the caption is fine too.

Line 310. "Contrarily" -- change to "In contrast"

Line 312. "aiming" -- change to "aimed"
Line 312. "shown" -- change to "show"

Line 332. "pix" -- change to "pixels"

Figure 10 caption.
"The vertical bar correspond to +/-1 the standard deviation..."
Change to
"The vertical bars correspond to +/-1 standard deviation..."

Line 392. Either "neural networks have" (if plural) or "neural network has" (if singular)

Line 399. "Discussion" (singular)

Line 405. "shows than" -- change to "shows that"

Line 409. "the larges" -- change to "the largest"

Line 415. "must include human SOMETHING" -- I suggest either "human judgment" or "human interaction"

Lines 419-420.
"still remains to be manual and may seem as the main hurdle"
I suggest
"still remains to be done manually and is the main hurdle"

Line 446. "mechanic" -- change to "mechanical"

Line 457. "out of scope" -- change to "outside the scope"
